# Effects of Five Dietary Carbohydrate Sources on Growth, Glucose Metabolism, Antioxidant Capacity and Immunity of Largemouth Bass (*Micropterus salmoides*)

**DOI:** 10.3390/ani14101492

**Published:** 2024-05-17

**Authors:** Pengcheng Qian, Yan Liu, Hao Zhang, Penghui Zhang, Yuanyuan Xie, Chenglong Wu

**Affiliations:** National-Local Joint Engineering Laboratory of Aquatic Animal Genetic Breeding and Nutrition (Zhejiang), Huzhou University, 759 East 2nd Road, Huzhou 313000, China; qianpc2019@outlook.com (P.Q.); zhlyyxxf@outlook.com (H.Z.); penghuizhang2021@outlook.com (P.Z.); xieyuanyuan23@outlook.com (Y.X.)

**Keywords:** *Micropterus salmoides*, carbohydrate sources, growth and hematological parameters, glucose metabolism, antioxidant and immune capability

## Abstract

**Simple Summary:**

Carbohydrate utilization is closely associated with level, chemical structure or processing method. Selecting an optimal carbohydrate source needs to take into consideration all these factors. This study investigated the effects of different carbohydrate sources, including monosaccharide and gelatinized or ungelatinized polysaccharide starch, on growth performance, hepatic glucose metabolism, antioxidant capacity, and immunity in juvenile largemouth bass. The results showed that dietary starch exhibited better growth performance independent of gelatinization. Ungelatinized potato starch was proved to be the more effective carbohydrate source for increasing hepatic glycolysis and ungelatinized tapioca starch better improved antioxidant and immune capacities. This study provides new information for selecting carbohydrate sources in artificial feed and helps improve fish health in farmed largemouth bass.

**Abstract:**

This study investigated the effects of glucose (GLU), tapioca starch (TS), gelatinized tapioca starch (GTS), potato starch (PS) and gelatinized potato starch (GPS) on growth and physiological responses in juvenile largemouth bass *Micropterus salmoides*. After 8 weeks, fish fed with starch diets had better weight gain and growth rates. Counts of red blood cells and monocytes were increased in the PS and GPS groups, compared to GLU group. Contents of serum triglyceride and total cholesterol were markedly elevated in the TS, PS and GPS groups. There were lower levels of serum glucose, insulin and cholecystokinin, and higher agouti-related peptide contents in the PS group compared to GLU group. PS and GPS could enhance glycolysis and TCA cycle by increasing their enzyme activities and transcriptional levels. Additionally, starch sources markedly heightened mRNA levels of key genes involved in the respiratory electron transport chain. Additionally, elevated mRNA levels of key antioxidant genes were shown in the TS and GTS groups. Moreover, TS and PS could promote immunity by upregulating transcriptional levels of the complement system, lysozyme and hepcidin. Taken together, starch exhibited better growth via increasing glycolysis and TCA cycle compared with GLU, and PS could improve antioxidant and immune capacities in largemouth bass.

## 1. Introduction

Carbohydrate is one of the most important energy sources and has been widely applied in aquatic animal diets due to its low cost. Increasing its proportion appropriately could ultimately spare protein for growth in diets [1,2,3]. Additionally, carbohydrate is used as important binders to increase the physical quality of feed pellets and help to shape particles [4]. The existing research reveals that the appropriate amount of carbohydrate promotes growth, feed utilization, digestive and metabolic function, and immune responses in fish [1,3]. The recommended carbohydrate addition is generally lower for carnivorous fish than for herbivorous and omnivorous fish in order to achieve these beneficial effects [1,5]. Conversely, excessive carbohydrate causes a series of negative effects, such as reduced growth, energy metabolism disturbance and defense system imbalance [2,5]. Therefore, the optimal carbohydrate content for the growth, metabolism, and immunity in farmed animals has long attracted the attention of many researchers.

Besides the supplementary level, the effects of dietary carbohydrate on fish health largely depend on its structure and processing technology [4,6,7]. Dietary carbohydrate can be categorized in three types, monosaccharide, disaccharide and polysaccharide, according to their chemical structure. In fact, starch polysaccharide is naturally the most widely distributed and the most common form of carbohydrate in diets [6]. In aquatic feeds, starch is an abundant source of polysaccharide, including tapioca starch, corn starch, potato starch, etc. These starch sources have lower amylose content and higher amylopectin content, which are also used as a better mucoadhesive polymer for producing feeds due to a high swelling power [8,9]. These attractive characteristics make them applied extensively in the feed industry. Health and growth-improving effects of tapioca and potato starch have been demonstrated in Barramundi *Lates calcarifer*, and Olive flounder *Paralichthys olivaceus* [10,11,12]. More recent studies have found that the utilization of carbohydrate sources in farmed animals varies with processing methods [8,13]. Among these processing parameters, the gelatinization degree has the greatest influence on carbohydrate absorption and digestion in animals [8]. Generally, gelatinized starch sources have higher digestibility compared to raw starch, because native starch after gelatinization treatment exhibits the functional properties, especially increasing metabolic activities in animals [13]. However, most studies in fish focus on one aspect of carbohydrate and comparatively study the impact of its level or molecular structure or gelatinization on growth and physiological responses in fish [10,14,15,16]. In fact, all aspects, including hematological variations, serum biochemical parameters and hormones, glucose metabolism, antioxidant and immune capabilities, need to be comprehensively considered to find an appropriate carbohydrate source for improving growth and health in different animals. Therefore, a further systematic study of the selection of carbohydrate in feeds is needed.

*Micropterus salmoides*, commonly known as largemouth bass, has a great economic value in the Chinese market. In recent years, the production of largemouth bass has expanded rapidly due to its fast growth rate, strong adaptability, and high nutritional value [17]. More high-quality compound feeds will be needed to meet the rising production of largemouth bass. However, nutritional requirements are still incomplete for this species. Although a certain amount of carbohydrate is required for the formation of pellets, further investigation is still necessary to determine which kind of carbohydrate sources are beneficial for improving growth, glucose metabolism and hepatic health of largemouth bass. Thus, this study was conducted to comprehensively investigate the physiological and metabolic responses of largemouth bass fed with five kinds of carbohydrate sources: monosaccharide (glucose) and polysaccharide (tapioca starch, gelatinized tapioca starch, potato starch and gelatinized potato starch). The optimal carbohydrate source in diets was selected by evaluating growth performance, serum biochemical parameters, glucose metabolism, antioxidant capacity and immune function. The results provide new insights for the development of artificial composite feeds for largemouth bass.

## 2. Materials and Methods

### 2.1. Animals and Diets Preparation

Juvenile largemouth bass were purchased from Deqing Longshengli Breeding farm (Huzhou, China). Before the feeding experiment, all fish were first acclimated to indoor culture settings and given a commercial diet in glass tanks with a capacity of 1000 L for a period of 2 weeks. Five isoprotein and isolipidic experimental diets were made at the National-Local Joint Engineering Laboratory of Aquatic Animal Genetic Breeding and Nutrition (Huzhou, China). Tapioca starch (TS), gelatinized tapioca starch (GTS), potato starch (PS), and gelatinized potato starch (GPS) were selected as starch sources. Glucose (GLU) was used as a control energy source in this study. The composition and nutrient levels of the experimental diets are shown in Table 1. The raw materials in diets were passed through a 60-mesh sieve and subsequently transformed into hard pellet feed with a diameter of 1.5 mm. Finally, the experimental diets were stored in the freezer at −20 °C.

A total of 375 fish (initial body weight: 10.0 ± 0.02 g) were randomly assigned to 5 groups in a closed circulating water system with triplicate allocations, each tank with a volume of 500 L containing 25 fish. All fish were provided with the experimental diets described above, which were administered at 3–4% of fish body weight twice daily (08:00 and 17:30), for 8 weeks. The temperature of the culture water was kept within the range of 26.5–28.5 °C, with a pH level between 6.8 and 7.3, and a dissolved oxygen concentration above 5.2 mg/L.

### 2.2. Sample Collection

After the feeding trial, all fish were fasted for 24 h. The fish were initially anesthetized according to the information outlined in a previous study [17]. They were then weighted and measured for subsequent growth parameters [15,16,17]. Blood samples were first taken from the tail vein of 15 fish for next blood cell identification and counting. Then, the remaining blood samples were centrifugated at 4 g °C and 3000× *g* for 10 min to collect serum samples and detect serum biochemical indexes. Livers of the 15 fish in each glass tank were collected, sampled and quick frozen in liquid nitrogen and stored at −80 °C for further use. Another 3 fish from each glass tank were chosen for whole body composition analysis and stored at −20 °C.

### 2.3. Serum Cell Counting and Biochemical Indexes

Blood cell identification and counting were conducted using a TEK 8500 VET automatic blood analyzer (Tekang Technology, Nanchang, China) following methods described by Wu et al. [18]. Metabolite contents and enzyme activities in the serum samples were detected on C400n automated hematology analyzer (Shenzhen, China) according to the corresponding instructions described by Yang et al. [17]. The contents variation of hormones modulating glucose metabolism and appetite in the serum samples were also measured using the commercial detecting kits (Jiancheng Biotech Co., Nanjing, China) following their corresponding instructions.

### 2.4. Determination of Hepatic Enzyme Activities about Glucose Metabolism

The 10 mM PBS solution (pH 7.4) was added to the liver samples and then homogenated in ice. After the homogenates were centrifuged at 3000× *g* for 20 min at 4 °C, the liquid supernatant samples were collected and kept at a temperature of −80 °C until it was ready to be examined. The quantification of protein was conducted using the Bradford method [19]. The activities of key enzymes related to glycolysis, tricarboxylic acid cycle (TCA), and glycogen synthesis in the liver samples were determined using diagnostic kits from Jiancheng Biotech Co. (Nanjing, China). The activities of glycogen phosphorylase (GPase) and glycogen branching enzyme (GBE) were assessed using enzyme-linked immunosorbent assay (ELISA) kits from Jiancheng Biotech Co. (Nanjing, China) and Comin Biotech Co. (Suzhou, China), respectively.

### 2.5. RNA Extraction and Analysis of Gene Expression

The total RNA in liver was extracted using Trizol reagent (Invitrogen, Carlsbad, CA, USA). Subsequently, cDNA samples were generated using the RT-PCR kit (Monad Biotech, Wuhan, China), following the protocol published previously [17]. Furthermore, all cDNA samples were kept at −80 °C for further real-time quantitative PCR detection and analysis. The primers used in the present study of genes involved in glucose metabolism, antioxidative response and immunity were listed in Table 2. β-actin and EF1α were both served as internal reference genes. Real-time PCR amplification was carried out in a volume of 25 μL with 12.5 μL of 2×SYBR Green Real-time PCR Master Mix (Takara, Dalian, China), 2 μL of cDNA mixture and 0.2 mmol each of primers. The real-time PCR temperature profile for the genes was 95 °C for 5 min followed by 35 cycles of 10 s at 95 °C, 15 s at TM (Table 2) and 40 s at 72 °C. After the amplification cycle of real-time PCR, the melting curves were systematically monitored (65 °C temperature gradient at 0.05 °C/10 s from 65 to 95 °C). Each sample was run in triplicate and the relative expression levels of the target genes were calculated using the 2^−ΔΔCT^ method [20].

### 2.6. Statistical Processing

The results were presented as mean ± standard deviation (SD). All data were first checked for data normality and homoscedasticity, and then underwent one-way analysis of variance (ANOVA) using SPSS 26.0. Tukey’s multiple range test was selected for the multiple comparisons among all the groups.

## 3. Results

### 3.1. Growth and Body Composition

The growth performance of largemouth bass fed different carbohydrate sources is shown in Table 3. Dietary carbohydrate sources did not affect SR (*p* > 0.05), but significantly changed the growth parameters of largemouth bass (*p* < 0.05). Fish had higher WG and SGR fed TS, GTS, PS and GPS diets, compared to the GLU diet (*p* < 0.05). However, the lower FCR was only observed in fish fed the GPS diet (*p* < 0.05). The lower VSI level was shown in the PS group compared to the GTS group (*p* < 0.05). Although there were no significant differences in the values of HSI among the GLU, GTS, and GPS groups (*p* > 0.05), the three groups had higher HSI than the TS and PS groups (*p* < 0.05). The GLU diet significantly decreased the CF value, which was lower than the other four diets (*p* < 0.05). The contents of moisture, crude lipid and ash in the whole body were comparable among the five groups (Table 3). However, the crude protein contents of the whole body in fish fed the TS and PS diets were significantly higher compared to the GLU diet (*p* < 0.05).

### 3.2. Blood Cell Indexes and Serum Biochemical Indices

The counts of WBC and MON were lower in the GLU group than those in the PS and GPS groups (*p* < 0.05) (Table 4). The lowest and the highest RBC counts were shown in the GLU group and the TS group, respectively (*p* < 0.05). Although intermediate RBC counts were mainly shown in the GTS, PS, and GPS groups, no marked differences were observed among the starch groups (*p* > 0.05). In addition, there were higher values of HGB and PLT and lower levels of MCV and NEU in the TS group compared to the GLU group (*p* < 0.05). There were no differences in the values of HGB and MCV among the GLU, GTS, PS and GPS groups (*p* > 0.05). The significantly lowest count of PLT was shown in the PS group compared to the TS, GTS, and GPS groups (*p* < 0.05). The lowest NEU counts were presented in the TS and GTS groups compared to the GLU, PS and GPS groups (*p* < 0.05). Moreover, the lowest LYM count and the highest EOS count were both presented in the GPS group compared to the GLU, TS, and GTS groups (*p* < 0.05). The values of HCT were not influenced by dietary carbohydrate sources (*p* > 0.05).

The levels of HDL-C and TG were markedly decreased in the serum of the GLU group, whereas their levels were significantly increased in the serum of the TS, GTS, PS and GPS groups (*p* < 0.05) (Table 5). In addition, dietary TS, GTS and PS significantly reduced the LDL-C contents, compared to dietary GLU (*p* < 0.05). TC contents were markedly increased in the TS, PS, and GPS groups compared to the GLU group (*p* < 0.05), and there were no significant differences between the GLU and GTS groups (*p* > 0.05). Dietary GLU increased the AST activities, compared to GTS, PS and GPS diets. Fish fed the PS diet had the lowest glucose level in serum than the other four diets, accompanied by a significant decrease in INS content (*p* < 0.05). The lowest GC content was shown in the GLU group compared to the TS, GTS, PS, and GPS groups (*p* < 0.05), although there were no significant differences in the GC levels among the four starch groups (*p* > 0.05). Dietary carbohydrate sources significantly affected levels of serum hormones involved in metabolic and appetite indices (*p* < 0.05). In GLU-fed fish, the concentrations of CCK were observably elevated, compared to the other four groups (*p* < 0.05). Conversely, the level of AGRP was largely reduced by the GLU diet (*p* < 0.05) (Table 5).

### 3.3. Hepatic Enzyme Activities Related to Glucose Metabolism

PS diet significantly increased the HK, PK and MDH activities, and the acetyl CoA content (*p* < 0.05) (Table 6). The effects of the GPS diet on the parameters above are similar to those of the PS diet, except that the values of HK and MDH were lower than the PS diet (*p* < 0.05). In contrast, fish fed the GLU diet displayed lower activities and content of HK, PFK, IDH and MDH, and the acetyl CoA. Dietary carbohydrate sources also changed glycogen contents and related enzyme activities. Fish fed the GLU and GPS diets exhibited higher glycogen levels than the TS diet (*p* < 0.05). Meanwhile, the GPS group had the highest GCS and GBE activities, and an intermediate GCS value was shown in the GLU group (*p* < 0.05). Glycogenolytic enzyme activity (GPa) is independent of carbohydrate sources in diets and no significant difference was found in the GPa among the five groups (*p* > 0.05) (Table 6).

### 3.4. Relative Expression of Genes Involved in Glucose Metabolism

Dietary PS significantly upregulated the expressions of genes including HK, MPC2, CS and MDH, while the increased expressions of IDH and PFK were found in the GTS diet (*p* < 0.05) (Figure 1). Furthermore, fish fed the GPS diet showed dramatically higher transcriptional levels of GYG1, GCS and GBE (*p* < 0.05) (Figure 2). The GTS group had a similar gene expression pattern in glycogen synthesis as the GPS group, whose expression were comparable between the two groups. Notably, the abilities of GLU diet to activate glycolysis, tricarboxylic acid cycle and glycogen synthesis was less than that of the other diets, which had lower mRNA levels of HK, PFK, PK, MPC2, PDHx, IDH, MDH, GYG1, GYG2, UGPa and GBE (*p* < 0.05) (Figure 1 and Figure 2).

### 3.5. Relative Expression of Genes in the Respiratory Electron Transport Chain

Polysaccharide-starch sources significantly heightened the expression variations of NDUfa1, ND2, SDHb, SDHc, and ATP6 compared with dietary GLU (*p* < 0.05) (Figure 3). In addition, the expression levels of ND1 were notably increased in fish fed with TS and PS diets, although there were no marked differences among the fish fed with GLU, GTS and GPS diets (*p* > 0.05). Higher levels of COX1 and COX3 were shown in the TS, GTS and PS groups compared with the GLU group (*p* < 0.05), while there were no differences between the GLU and GPS groups (*p* > 0.05) (Figure 3).

### 3.6. Relative Expression of Genes Involved in Antioxidant and Immune Capability

The expressions of antioxidant enzyme genes including Cu/Zn-SOD, Mn-SOD, CAT, GPX1b, GPX7, and GR in the TS and GTS groups were markedly elevated, compared to the GLU group (*p* < 0.05) (Figure 4). There was a similar trend in the expressions of the genes, including Cu/Zn-SOD, Mn-SOD, CAT and GPX1b in the PS group. Higher mRNA levels of Nrf2 were found in the PS group compared to the GLU and GPS groups (*p* < 0.05), while there were higher mRNA levels of Keap1b in the GLU group compared to the GPS group (*p* < 0.05). Notably, dietary GTS and GPS remarkedly upregulated the expression of GSS (*p* < 0.05). Additionally, the significantly higher mRNA levels of C3, C4, C8a, C8b, C8g, LYZ and Hepc were observed in the TS and PS groups compared to the GLU group (*p* < 0.05) (Figure 5). Additionally, no significant differences were found in the C3, C8a, C8b, LZM, and Hepc mRNA levels between the GLU and GPS groups (*p* > 0.05). The transcriptional levels of all immune genes in the TS group were significantly increased compared with the GTS groups (*p* < 0.05). A similar result was also shown between the PS and the GPS groups (Figure 5).

## 4. Discussion

An appropriate level of carbohydrate in feed can promote fish growth and save protein by transferring amino acids from oxidative pathways [21,22]. Previous studies have found the impact of different carbohydrate sources on growth performance varies widely depending on their structure [14]. In this study, the WG, SGR, and CF of largemouth bass juveniles fed with GLU were lower compared to other groups, indicating a poorer utilization capability of glucose. That is likely because glucose can be rapidly absorbed by fish, leading to a prolonged spike in blood glucose levels, which, in turn, results in decreased growth [14,23,24,25]. In contrast, largemouth bass fed with polysaccharide-starch sources obtained better growth efficiency and higher crude protein levels compared to GLU group in this experiment, indicating a stronger carbohydrate utilization and protein-sparing capability [14,25]. Similar results to this experiment were found in studies involving species such as blunt snout bream *Megalobrama amblycephala* [26], Nile tilapia *Oreochromis niloticus* [25], golden pompano *Trachinotus ovatus* [27], and swamp eel *Monopterus albus* [28]. These previous findings and our results indicated that largemouth bass could exhibit better utilization of structurally complex carbohydrate sources [29].

Compared with the GLU group, the polysaccharide groups resulted in higher HDL-C and TG contents, and lower LDL-C contents, aligned with previous findings in white sturgeon *Acipenser transmontanus* [30] and large yellow croaker *Pseudosciaena crocea* R. [31], which were due to increased active endogenous lipid transport induced. Many studies have proven that higher serum AST activity relates with liver damage and poor liver health [32,33]. In this study, AST activities were heightened in the serum of the GLU group, indicating that dietary GLU could impair liver health in largemouth bass to some extent. Moreover, endogenous hormones always play crucially regulatory functions in glucose metabolic process in fish species [17,34,35]. Generally, serum glucose level can be reduced by INS via increasing glucose uptake, while increased by GC via elevating glucose production in response to various external nutrients [36,37,38]. Therefore, higher GC and lower INS contents were also shown in the PS and GPS groups, which might respond to variations of corresponding serum glucose contents [35,38]. In addition, many studies have reported higher levels of AGRP could increase the appetite in animals [17,35,39]. Considering the lower FCR in the GTS and GPS groups, it is suggested that gelatinized polysaccharide (GTS and GPS) could improve feed utilization and subsequently enhance growth by increasing AGRP contents in largemouth bass.

Various carbohydrate sources can exert different effects on glucose metabolic capabilities by regulating these crucial enzymes (HK, PK, PFK, IDH, and MDH) involved in glycolysis and the TCA cycle in animals. The current investigation revealed that polysaccharide-starch sources had notable impacts on the variations in enzyme activities and mRNA expression levels of HK, PK, PFK, IDH, and MDH in largemouth bass, which was consistent with previous studies in giant grouper *Epinephelus lanceolatus* larvae [24], Nile tilapia *O. niloticus* [25] and blunt snout bream *M. amblycephala* [40]. In general, MPC2 transports cytoplasmic pyruvate into mitochondria, and subsequently, PDH converts pyruvate to acetyl CoA for further metabolism through the TCA cycle. Our results showed that higher expression levels of MPC2 and PDH, as well as higher contents of pyruvate and acetyl CoA, were observed in the TS, GTS, PS, and GPS groups, compared with GLU group, which was in line with results in Nile tilapia *O. niloticus* [25]. It indicated that polysaccharide-starch sources could notably improve carbohydrate utilization via activating glycolysis and TCA cycle in largemouth bass. In addition, higher hepatic glycogen contents were shown in the GLU group compared with the TS group, consistent with findings in Amur sturgeon *Acipenser schrenckii* [41]. Meanwhile, slightly higher glycogen contents were also found in the GTS and GPS groups compared to the TS and PS groups, respectively, which was similar to results in European sea bass *Dicentrarchus labrax* juveniles [42], indicating gelatinized starch could induce glycogen synthesis in largemouth bass. Moreover, as key regulating molecules in the glycogen synthesizing system, the levels of GYG1, GCS, and GBE were also influenced by dietary carbohydrate sources, exhibiting similar variation trends to those of glycogen contents. Higher expression of GYG1, GCS, and GBE in the GTS and GPS groups also verified that gelatinized starch had more ability to synthesize glycogen that partly explained why they could be better utilized in the diet as energy sources. However, some reports also found glycogen contents were not changed, independently of the degree of starch gelatinization in blunt snout bream *M. amblycephala* [43]. These differences may be attributed to different species of fish. Given these findings and growth indices, the results suggested that gelatinized starch could improve carbohydrate utilization by activating glycogen synthesis in largemouth bass.

It is well-known that adenosine triphosphate (ATP) synthesis is primarily generated during the processes of mitochondrial oxidative phosphorylation (OXPHOS) in eukaryotic cells [44]. OXPHOS is carried out by mitochondrial respiratory electron transport chain (ETC) during nutritional metabolic processes of different nutrients, including carbohydrate sources [45]. Our study found that the expressions of key functional genes, such as NDUfa1, ND2, SDHb, SDHc, and ATP6 in ETC, were significantly increased in polysaccharide-starch (TS, GTS, PS, and GPS) groups compared to the GLU group. These findings suggested that polysaccharide sources could activate the respiratory chain complexes and enhance oxidative phosphorylation by increasing the expression levels of these key functional genes in the ETC through transcriptional modifications in largemouth bass [43,45]. Meanwhile, ROS is generated during the activation of respiratory chain complexes and oxidative phosphorylation [46]. In order to alleviate the potential oxidative stress and maintain redox homeostasis, aerobic organisms have evolved numerous antioxidant enzymes, including SOD, CAT, GSS, GPX, and GR, etc. [32,46]. In this study, we observed markedly elevated levels of Cu/Zn-SOD, Mn-SOD, and GPX1b in the four starch groups compared to the GLU group. This indicated that polysaccharide sources enhanced antioxidant capabilities in largemouth bass. Furthermore, the expression levels of Cu/Zn-SOD, Mn-SOD, GPX1b, and GR were higher in the TS and GTS groups than in the PS and GPS groups, suggesting that TS and GTS may be more effective at improving antioxidant status in largemouth bass. Moreover, our results also found higher Nrf2 levels and lower Keap1b levels were mainly shown in the PS group and the GPS group compared to the GLU group, respectively, consistent with previous findings in fish species with better antioxidant capability [32,47]. Given the pivotal roles of Nrf2/Keap1 in initiating the expression of antioxidant enzymes, it demonstrated that adequate polysaccharide sources could contribute to maintaining redox homeostasis and improve antioxidant capability in largemouth bass.

The complement system plays an essential role in both innate and adaptive immune responses, serving to alert the host of potential pathogens and assist in their removal [46]. Increasing studies on fish have verified that the response of the complement system is strongly correlated with dietary carbohydrate sources [48,49]. In the current study, we found that polysaccharide carbohydrate sources more effectively initiated immune function compared to monosaccharide, and that ungelatinized starch was superior to gelatinized starch in this respect. For instance, the TS diet significantly increased C3 and C4 mRNA levels, indicating that the complement pathway was activated, thus enabling a beneficial response to inflammation [49]. Additionally, activated C3 and/or C4 are known to assist in the recognition and phagocytosis of microbes by phagocytes such as neutrophils, macrophages and monocytes [50,51]. While the TS group in Table 4 showed a lower number of NEU and MON, the increased C3 and C4 levels likely facilitated the bactericidal effect of these phagocytes in the blood by enhancing their phagocytic capacity, rather than by increasing the number of blood cells. The higher value of RBC in Table 4 suggested that TS-fed fish had a very robust immune defense system, compared to those fed other starch sources, in line with a previous study in cobia (*Rachycentron canadum Linnaeus.*) [14]. Moreover, upregulation of C8a, C8b and C8g (downstream gene of C3 and C4) in TS-fed fish further promoted the formation of the membrane attack complex, which led to cytolysis of pathogens. The findings indicated that tapioca starch provided a higher level of immunity through accelerating complement-mediated bactericidal and lytic effects in fish. Lysozyme, another key factor of innate immunity, causes the lysis of bacterial walls and is often used to assess the immune status in aquatic animals [52]. Our results showed that fish fed with the TS diet had higher activities of LZM at the transcriptional level. This demonstrated again that the TS diet exhibited a greater ability to improve the innate immunity of largemouth bass. Data previously proved that ungelatinized starch had a significant effect on activating the immune function [12]. In this study, the mRNA levels of all the genes above described in the GTS group were lower than that of the TS group, in agreement with the results of previous research [10,11]. Antimicrobial peptides such as Hepc also participate in fish innate immunity, produced primarily in the liver [53,54]. A growing body of studies has reported that prior to infection and after challenged by pathogenic bacteria (*Aeromonas hydrophila* and *Pseudomonas fluorescens*), Hepc can down-regulate pro-inflammatory markers or directly kill bacteria to protect fish against inflammation and bacterial infection [55,56,57,58,59,60]. Our study found that Hepc expression was increased induced by dietary TS, followed by PS and GPS, indicating better immunomodulatory effects. In short, our results revealed that ungelatinized tapioca starch has a more positive effect on the immunity of largemouth bass.

## 5. Conclusions

In summary, compared with GLU, dietary starch sources (TS, GTS, PS and GPS) exhibited better growth performance and increased the crude protein content of the whole body in largemouth bass. Meanwhile, dietary starch sources (TS, GTS, PS and GPS) improved carbohydrate-utilizing capability via activating glycolysis, TCA cycle, mitochondrial oxidative phosphorylation and glycogen synthesis in largemouth bass. In addition, antioxidant abilities were significantly elevated in the TS and GTS groups, and immune responses were notably promoted in the TS and PS groups compared with the GLU group. This study provides new information for selecting carbohydrate sources in artificial feed and helps improve fish health in farmed largemouth bass.

## Figures and Tables

**Figure 1 animals-14-01492-f001:**
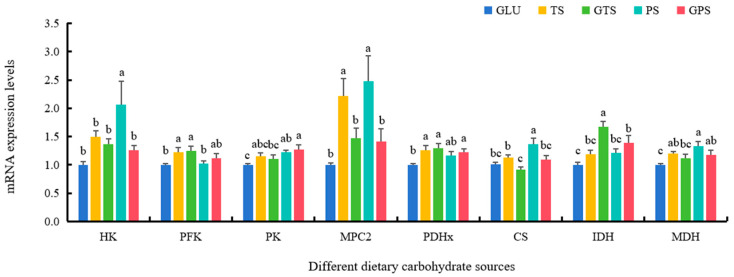
Effect of dietary carbohydrate sources on mRNA expression levels of genes involved in the glycolysis and citric acid cycle in largemouth bass (*n* = 3). Bars with different superscripts are significantly different (*p* < 0.05). HK, hexokinase-1; PFK, phosphofructokinase; PK, pyruvate kinase; MPC2, mitochondrial pyruvate carrier 2; PDHx, pyruvate dehydrogenase complex component X; CS, citrate synthase; IDH, isocitrate dehydrogenase; MDH, malate dehydrogenase.

**Figure 2 animals-14-01492-f002:**
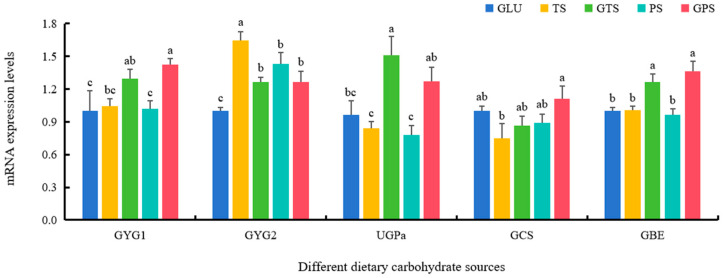
Effect of dietary carbohydrate sources on mRNA expression levels of genes involved in the glycogen synthesis in largemouth bass (*n* = 3). Bars with different superscripts are significantly different (*p* < 0.05). GYG1, glycogenin-1; GYG2, glycogenin-2; UGPα, UTP–glucose-1-phosphate uridylyltransferase; GCS, glycogen synthase; GBE, glycogen branching enzyme.

**Figure 3 animals-14-01492-f003:**
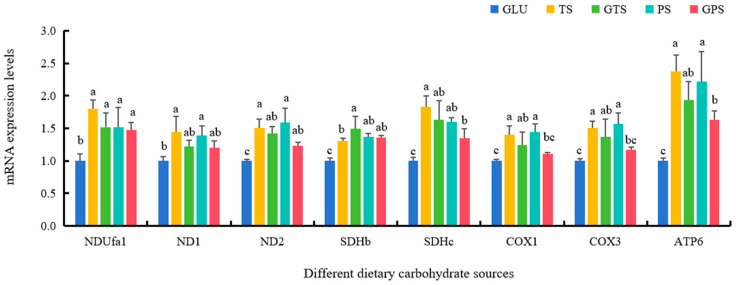
Effect of dietary carbohydrate sources on mRNA expression levels of mitochondrial respiratory electron transport chain genes in largemouth bass (*n* = 3). Bars with different superscripts are significantly different (*p* < 0.05). NDUfa1, NADH:ubiquinone oxidoreductase subunit A1; ND1, NADH dehydrogenase subunit 1; ND2, NADH dehydrogenase subunit 2; SDHb, succinate dehydrogenase complex iron sulfur subunit B; SDHc, succinate dehydrogenase complex iron sulfur subunit C; COX1, cytochrome c oxidase I; COX3, cytochrome c oxidase III; ATP6, ATP synthase 6.

**Figure 4 animals-14-01492-f004:**
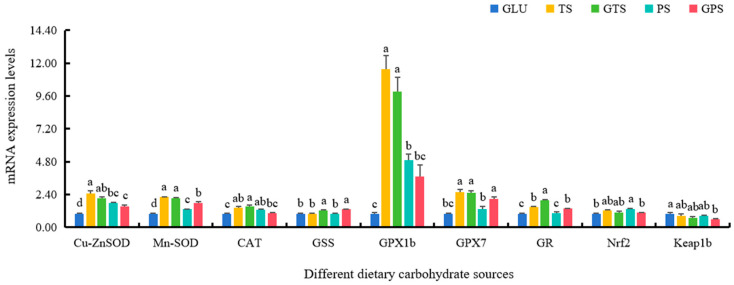
Effect of dietary carbohydrate sources on mRNA expression levels of antioxidant-relative genes in largemouth bass (*n* = 3). Bars with different superscripts are significantly different (*p* < 0.05). Cu/Zn-SOD, Cu/Zn-Superoxide dismutase; Mn-SOD, Mn-Superoxide dismutase; CAT, catalase; GSS, glutathione synthase; GR, glutathione reductase; GPX1b, glutathione peroxidase 1b; GPX7, glutathione peroxidase 7; Nrf2, nuclear factor E2-related factor 2; keap1b, Kelch-like ECH associated protein 1b.

**Figure 5 animals-14-01492-f005:**
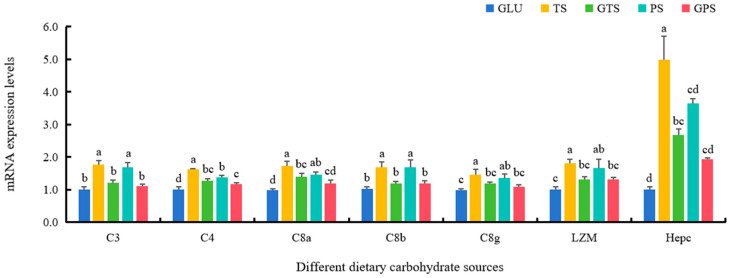
Effect of dietary carbohydrate sources on mRNA expression levels of immune genes in largemouth bass (*n* = 3). Bars with different superscripts are significantly different (*p* < 0.05). C3, complement component 3; C4, complement component 4; C8a, complement component 8a; C8b, complement component 8b; C8g, complement component 8g; C9, complement component 9; LZM, lysozyme; Hepc, hepcidin; β-actin, actin beta.

**Table 1 animals-14-01492-t001:** Composition and nutrient levels of experimental diets (% dry matter).

Ingredients	GLU	TS	GTS	PS	GPS
Casein ^a^	27.0	27.0	27.0	27.0	27.0
Fish meal ^b^	37.0	37.0	37.0	37.0	37.0
Soybean oil ^c^	7.0	7.0	7.0	7.0	7.0
Lecithin ^b^	2.0	2.0	2.0	2.0	2.0
Mineral premix ^d^	2.5	2.5	2.5	2.5	2.5
Vitamin premix ^e^	1.2	1.2	1.2	1.2	1.2
Choline chloride ^a^	0.4	0.4	0.4	0.4	0.4
Microcrystalline cellulose ^f^	12.9	12.9	12.9	12.9	12.9
Glucose (GLU) ^a^	10.0				
Tapioca starch (TS) ^g^		10.0			
Gelatinized tapioca starch (GTS) ^g^			10.0		
Potato starch (PS) ^g^				10.0	
Gelatinized potato starch (GPS) ^g^					10.0
Total	100.0	100.0	100.0	100.0	100.0
Proximate composition					
Crude protein (%)	48.98	48.68	48.72	48.81	48.85
Crude lipid (%)	12.07	12.15	12.21	12.19	12.12
Ash (%)	4.91	4.91	4.93	4.92	4.92

^a^ Obtained from Sinopharm Chemical Reagent Co., Ltd., Shanghai, China; Casein crude protein 80.56%. ^b^ Obtained from Zhejiang Dongyu Biotechnology Co., Ltd., Huzhou, China; Fish meal crude protein 67.6%. ^c^ Soybean oil, obtained from COFCO Group Co., Ltd., Beijing, China. ^d^ Mineral premix (mg/kg): KI, 0.4 mg; CoCl_2_·6H_2_O, 52 mg; CuSO_4_·5H_2_O, 16 mg; FeSO_4_·7H_2_O, 200 mg; ZnSO_4_·H_2_O, 280 mg; MnSO_4_·H_2_O, 45 mg; MgSO_4_·7H_2_O, 1200 mg; Ca(H_2_PO_4_)_2_, 12,000 mg; NaCl, 60 mg. ^e^ Vitamin premix (mg/kg): vitamin A, 20 mg; vitamin D_3_, 3 mg; vitamin C, 300 mg; vitamin E, 300 mg; thiamin, 20 mg; riboflavin, 10 mg; pyridoxine HCl, 20 mg; vitamin B_12_, 0.2 mg; vitamin K_3_, 5 mg; inositol, 1000 mg; pantothenic acid, 30 mg; folic acid, 3 mg; niacin acid, 50 mg; biotin, 1 mg. ^f^ Obtained from Linghu Chemical Co., Ltd., Huzhou, China. ^g^ Obtained from Jiaxing Xinxin Food Technology Co., Ltd., Jiaxing, China.

**Table 2 animals-14-01492-t002:** Primers for real-time qPCR used in this study.

Gene	Forward Primer (5′-3′)	Reverse Primer (5′-3′)	Gene ID	TM (°C)	E-Values (%)
HK	GGCTTCACCTTCTCATTTCC	CCTCAACAGTCCCACCACA	119889994	57.5	101.6
PFK	CTGTATAATCCCTGCCACCAT	CACCACAAACACTCGCCTC	119904003	58.0	99.8
PK	CACCAACCCATTCATTTGCA	AGTGTCATCACCTCAGAGTAGCG	119897542	59.5	102.7
MPC2	ACTGGCCTGATCTGGTCC	TGCCTTCATATCCTGCTTG	119904262	57.5	98.6
PDHx	AAGATGCGTTGAGTCTGTTGAA	TTGGGTCTGCTGCCTGTG	119893818	58.5	101.5
CS	ACTCAAGTCGGGAAGGGTT	CAGGTGCTTCAGGGCAAA	119905193	57.5	99.7
IDH	ACCAATCCCATAGCCTCCATC	TCCAGTGCCTCAGCGAACA	119904575	60.0	104.8
MDH	GCTTCACCTTCTCCGTCCTG	GGTTCTTCTCAATGCCGTTC	119897930	59.5	104.2
GYG1	GGAAACTGGTCGCACTCG	GGGACGCTTCATCAGGGA	119910845	58.5	101.6
GYG2	TCCGTTCTAAGATGCTTCATTC	TCAACTCCTCTGCTCCAAAA	119905587	57.5	98.8
UGPα	GAGGAGTCGATCCAGCCCTAT	CCAGACCGCCGTTCAGTT	119915305	59.5	102.3
GCS	TGTCATCGGTCATTTCCACG	CAGCACAGAGGTATCGTCCC	119918252	59.5	103.5
GBE	ACGACTGGGTTCACTGGG	TGTATGAGGCGACCTTTCC	119883477	57.0	101.4
NDUfa1	ATGTGGTATGAAATTCTGCCTAG	CTCACCTTCCCTCCGTTG	119888751	57.5	98.6
ND1	CTCCCACATTCCCACGAT	GCATGAGCTGGTCATAGCG	4100035	57.0	99.4
ND2	AAATACCCTAGCCATCATCCC	CCTGTGAGCCCAGCGTTA	4100026	58.0	102.7
SDHb	GCCGCTGTGGTGTTGTG	TCTCCTGGAGTGTCTGGGTC	119886220	57.5	103.5
SDHc	GGCATCGCCTTTCCGTTAT	CGGTGCGGTAGAGTTCTGG	119901747	60.0	101.8
COX1	CCGAAACCTCAACACCACC	TGAAATCATCCCGAACCCT	4100027	59.0	99.7
COX3	CATTATCGGCTCAACTTTCCT	TCAATATCATGCTGCTGCTTC	4100031	57.5	102.4
ATP6	CACCCTCCTAATCCCTGTTC	GAGATGTCCGGCTGTCAAA	4100030	56.5	100.8
Cu/Zn-SOD	TGAGCAGGAGGGCGATTC	GCACTGATGCACCCATTTGTA	119895678	59.5	102.7
Mn-SOD	CAGGGATCTACAGGTCTCATT	ACGCTCGCTCACATTCTC	119909030	55.0	98.5
CAT	ACCTATTGCTGTCCGCTTCTC	TCCCAGTTGCCCTCCTCA	119893048	59.5	104.8
GSS	ACAGGAGCAAAGCAAGCA	TTCCACCAAGAACATGACG	119905734	56.0	100.7
GR	ACGCCATCACGAGCAGG	CATCTCATCACAGCCCAAGC	119889417	59.0	103.2
GPX1b	CAACCAGTTCGGGCATCA	GGCATCCTTCCCATTCACA	119886925	59.5	101.8
GPX7	ACCAAGTCTCCTTCCCTCTG	CCAATCAGGCTCCTTTCC	119900239	56.0	99.7
Nrf2	CAAAGACAAGCGTAAGAAGC	CAGGCAGATTGATAATCATAGA	119904119	55.0	98.6
keap1b	CCTTACTCCAGGCTGTCCG	GAAATTACTTTGGTGGGTTTGT	119899192	58.5	103.1
C3	CGAGACCTGCTCCATCCTA	TCTGGGTGAGTCGGTGCTT	119899478	59.5	101.7
C4	TGCTCGCACCCGAAACA	CAGCCTCAAATCCACTCAGAAG	119897763	60.0	102.9
C8a	TGGTGGCACAGAGTGTATTG	GTTTCTTTGCAGGTGAAGCT	119902403	56.0	98.4
C8b	AGGCAGGAGGTGGAAGAGTA	AGCAGCCGCCAGCGTAAT	119902405	60.5	99.3
C8g	CAGTGGTTGCGTTGATGTG	GTCTATGTTCTGTGCGGGTG	119908612	57.0	100.3
LZM	TGTCCAAGTGGGAGTCAGG	GTTGCATCCATTCGCAGA	119914057	56.5	102.6
Hepc	CTCTGCCGTCCCATTCAC	GCATCATCCACGATTCCATT	119897237	58.0	101.9
β-actin	TCCTGCGTCTTGACTTGG	GATTTCCCTTTCGGCTGT	119885147	58.7	103.8
EF1α	CGTCAAGGAAATCCGTCGTG	GCGTAACCTGCGTTGATCTGG	119907150	59.5	102.3

HK, hexokinase-1; PFK, phosphofructokinase; PK, pyruvate kinase; MPC2, mitochondrial pyruvate carrier 2; PDHx, pyruvate dehydrogenase complex component X; CS, citrate synthase; IDH, isocitrate dehydrogenase; MDH, malate dehydrogenase; GYG1, glycogenin-1; GYG2, glycogenin-2; UGPα, UTP-glucose-1-phosphate uridylyl-transferase; GCS, glycogen synthase; GBE, glycogen branching enzyme; NDUfa1, NADH:ubiquinone oxidoreductase subunit A1; ND1, NADH dehydrogenase subunit 1; ND2, NADH dehydrogenase subunit 2; SDHb, succinate dehydrogenase complex iron sulfur subunit B; SDHc, succinate dehydrogenase complex iron sulfur subunit C; COX1, cytochrome c oxidase I; COX3, cytochrome c oxidase III; ATP6, ATP synthase 6; Cu/Zn-SOD, Cu/Zn-Superoxide dismutase; Mn-SOD, Mn-Superoxide dismutase; CAT, catalase; GSS, glutathione synthase; GR, glutathione reductase; GPX1b, glutathione peroxidase 1b; GPX7, glutathione peroxidase 7; Nrf2, nuclear factor E2-related factor 2; keap1b, Kelch-like ECH associated protein 1b; C3, complement component 3; C4, complement component 4; C8a, complement component 8a; C8b, complement component 8b; C8g, complement component 8g; LZM, lysozyme; Hepc, hepcidin; β-actin, actin beta; EF1α, elongation factor-1α.

**Table 3 animals-14-01492-t003:** Effects of dietary carbohydrate sources on growth performance and whole body composition of largemouth bass (*n* = 3). Means in each row with different superscripts show significant differences (*p* < 0.05).

Items	GLU	TS	GTS	PS	GPS
IBW/(g)	9.99 ± 0.01	10.00 ± 0.00	9.98 ± 0.02	10.02 ± 0.01	9.99 ± 0.01
FBW/(g)	59.83 ± 0.54 ^b^	64.45 ± 0.21 ^a^	67.36 ± 0.62 ^a^	66.19 ± 1.51 ^a^	67.39 ± 0.88 ^a^
WG/(%)	498.78 ± 4.70 ^b^	544.84 ± 5.43 ^a^	574.84 ± 6.90 ^a^	560.62 ± 14.56 ^a^	574.60 ± 9.15 ^a^
SGR/(%/d)	6.74 ± 0.02 ^b^	6.89 ± 0.05 ^a^	6.98 ± 0.02 ^a^	6.94 ± 0.05 ^a^	6.98 ± 0.03 ^a^
FCR	1.00 ± 0.02 ^a^	0.98 ± 0.00 ^a^	0.92 ± 0.01 ^ab^	0.97 ± 0.01 ^a^	0.86 ± 0.02 ^b^
VSI/(%)	9.15 ± 0.23 ^ab^	9.15 ± 0.25 ^ab^	9.67 ± 0.18 ^a^	8.64 ± 0.10 ^b^	9.22 ± 0.09 ^ab^
HSI/(%)	1.93 ± 0.06 ^a^	1.31 ± 0.12 ^b^	1.92 ± 0.01 ^a^	1.12 ± 0.03 ^b^	1.80 ± 0.09 ^a^
CF/(g/cm^3^)	2.00 ± 0.01 ^b^	2.21 ± 0.03 ^a^	2.24 ± 0.05 ^a^	2.29 ± 0.05 ^a^	2.33 ± 0.05 ^a^
SR/(%)	94.67 ± 1.33	96.00 ± 4.62	96.33 ± 3.53	96.00 ± 3.25	96.00 ± 4.00
Body composition (%)					
Moisture	72.19 ± 0.17	72.52 ± 0.17	72.60 ± 0.21	72.36 ± 0.18	72.41 ± 0.22
Crude protein	16.02 ± 0.04 ^b^	16.96 ± 0.06 ^a^	16.63 ± 0.14 ^ab^	16.82 ± 0.15 ^a^	16.50 ± 0.31 ^ab^
Crude lipid	7.58 ± 0.09	7.34 ± 0.07	7.55 ± 0.17	7.20 ± 0.03	7.26 ± 0.05
Ash	3.65 ± 0.03	3.79 ± 0.05	3.75 ± 0.06	3.68 ± 0.04	3.67 ± 0.02

IBW, initial body weight; FBW, final body weight; WG, weight gain; SGR, specific growth rate; FCR, feed conversion ratio; VSI, viscerosomatic index; HSI, hepatosomatic index; CF, condition factor; SR, survival ratio.

**Table 4 animals-14-01492-t004:** Effects of dietary carbohydrate sources on blood cell indexes of largemouth bass (*n* = 12). Means in each row with different superscripts show significant differences (*p* < 0.05).

Items	GLU	TS	GTS	PS	GPS
WBC/(10^9^/L)	175.97 ± 2.03 ^c^	180.00 ± 0.62 ^bc^	175.42 ± 0.60 ^c^	183.05 ± 1.17 ^b^	189.28 ± 0.76 ^a^
RBC/(10^12^/L)	2.13 ± 0.03 ^c^	2.53 ± 0.01 ^a^	2.28 ± 0.01 ^b^	2.32 ± 0.01 ^b^	2.31 ± 0.02 ^b^
HGB/(g/L)	75.33 ± 1.33 ^b^	82.00 ± 0.58 ^a^	74.67 ± 0.67 ^b^	78.33 ± 0.33 ^ab^	76.33 ± 0.67 ^b^
MCV/(fL)	152.07 ± 0.87 ^a^	144.53 ± 0.68 ^b^	149.30 ± 0.56 ^a^	152.07 ± 0.37 ^a^	151.33 ± 0.78 ^a^
PLT/(10^9^/L)	56.67 ± 0.88 ^bc^	63.33 ± 1.45 ^a^	60.00 ± 1.53 ^ab^	53.67 ± 1.45 ^c^	59.67 ± 0.88 ^ab^
NEU/(10^9^/L)	12.00 ± 0.10 ^ab^	9.50 ± 0.15 ^c^	9.63 ± 0.13 ^c^	11.30 ± 0.15 ^b^	12.17 ± 0.23 ^a^
LYM/(10^9^/L)	78.33 ± 1.07 ^a^	77.33 ± 0.23 ^ab^	77.67 ± 0.88 ^ab^	74.97 ± 0.13 ^bc^	72.73 ± 0.43 ^c^
MON/(10^9^/L)	11.53 ± 0.12 ^c^	12.63 ± 0.12 ^bc^	12.47 ± 0.32 ^bc^	13.47 ± 0.32 ^ab^	13.97 ± 0.33 ^a^
EOS/(10^9^/L)	0.10 ± 0.00 ^b^	0.10 ± 0.00 ^b^	0.10 ± 0.00 ^b^	0.10 ± 0.00 ^b^	0.33 ± 0.37 ^a^
HCT/(%)	32.67 ± 0.49	34.63 ± 0.62	33.70 ± 0.29	33.77 ± 0.86	34.97 ± 0.35

WBC, white blood cell; RBC, red blood cell; HGB, hemoglobin; MCV, mean corpuscular volume; PLT, platelet; NEU, neutrophil; LYM, lymphocyte; MON, monocyte; EOS, eosinophils; HCT, hematocrit.

**Table 5 animals-14-01492-t005:** Effects of dietary carbohydrate sources on serum biochemical indices and hormones of largemouth bass (*n* = 6). Means in each row with different superscripts show significant differences (*p* < 0.05).

Items	GLU	TS	GTS	PS	GPS
HDL-C/(mmol/L)	1.33 ± 0.01 ^d^	1.68 ± 0.02 ^a^	1.47 ± 0.02 ^c^	1.54 ± 0.04 ^bc^	1.62 ± 0.04 ^ab^
LDL-C/(mmol/L)	1.53 ± 0.04 ^a^	1.27 ± 0.02 ^c^	1.37 ± 0.01 ^bc^	1.33 ± 0.03 ^bc^	1.47 ± 0.05 ^ab^
TG/(mmol/L)	5.71 ± 0.25 ^b^	9.90 ± 1.85 ^a^	9.87 ± 0.24 ^a^	10.70 ± 0.20 ^a^	10.53 ± 0.41 ^a^
TC/(mmol/L)	4.88 ± 0.01 ^c^	5.87 ± 0.27 ^ab^	5.42 ± 0.22 ^bc^	6.41 ± 0.08 ^a^	6.31 ± 0.03 ^a^
AST/(U/L)	44.42 ± 2.98 ^a^	39.07 ± 3.09 ^ab^	21.90 ± 0.40 ^c^	21.73 ± 0.65 ^c^	30.37 ± 1.53 ^bc^
GLU/(mmol/L)	6.36 ± 0.01 ^b^	6.62 ± 0.03 ^b^	7.66 ± 0.26 ^a^	4.42 ± 0.06 ^d^	5.49 ± 0.24 ^c^
GC/(ng/L)	248.02 ± 10.40 ^b^	359.17 ± 10.16 ^a^	361.74 ± 19.16 ^a^	324.90 ± 17.64 ^a^	363.22 ± 9.85 ^a^
INS/(mIU/L)	8.69 ± 0.15 ^a^	7.56 ± 0.16 ^ab^	8.01 ± 0.42 ^a^	5.73 ± 0.55 ^c^	6.23 ± 0.23 ^bc^
CCK/(pg/mL)	230.76 ± 9.94 ^a^	117.44 ± 1.30 ^b^	123.60 ± 6.42 ^b^	104.85 ± 1.70 ^b^	109.07 ± 11.33 ^b^
AGRP/(mg/L)	10.94 ± 2.22 ^c^	36.18 ± 1.88 ^ab^	38.24 ± 0.78 ^a^	28.33 ± 0.75 ^b^	36.46 ± 2.50 ^ab^

HDL-C, high-density lipoprotein cholesterol; LDL-C, low-density lipoprotein cholesterol; TG, triglycerides; TC, total cholesterol; AST, aspartate aminotransferase; GLU, glucose; GC, glucagon; INS, insulin; CCK, cholecystokinin; AGRP, agouti-related peptide.

**Table 6 animals-14-01492-t006:** Effects of dietary carbohydrate sources on glucose metabolic parameters in the liver of largemouth bass (*n* = 3). Means in each row with different superscripts show significant differences (*p* < 0.05).

Items	GLU	TS	GTS	PS	GPS
HK/(nmol/min/mg prot)	0.27 ± 0.02 ^c^	0.36 ± 0.01 ^b^	0.34 ± 0.01 ^b^	0.46 ± 0.00 ^a^	0.33 ± 0.01 ^b^
PFK/(U/mg prot)	3.22 ± 0.07 ^b^	3.59 ± 0.10 ^ab^	3.83 ± 0.08 ^a^	3.33 ± 0.08 ^b^	3.30 ± 0.17 ^b^
PK/(U/mg prot)	124.24 ± 3.70 ^b^	137.47 ± 2.26 ^b^	154.55 ± 4.52 ^b^	344.11 ± 26.30 ^a^	350.09 ± 19.30 ^a^
pyruvate/(U/mg prot)	2.91 ± 0.13	3.45 ± 0.19	3.11 ± 0.25	3.46 ± 0.14	3.35 ± 0.08
Acetyl CoA/(U/g prot)	1171.28 ± 23.21 ^b^	1232.30 ± 26.05 ^ab^	1178.46 ± 22.51 ^b^	1277.18 ± 10.29 ^a^	1290.00 ± 14.56 ^a^
LDH/(U/g prot)	23.44 ± 0.58	26.30 ± 4.33	17.69 ± 0.54	21.26 ± 0.23	22.92 ± 0.28
IDH/(nmol/min/mg prot)	113.38 ± 2.51 ^c^	125.03 ± 2.97 ^bc^	153.50 ± 7.41 ^a^	127.16 ± 2.35 ^bc^	138.35 ± 5.17 ^ab^
MDH/(U/mg prot)	8.25 ± 0.11 ^c^	12.07 ± 0.60 ^b^	10.90 ± 0.16 ^b^	13.59 ± 0.35 ^a^	11.43 ± 0.10 ^b^
GPa/(nmol/min/mg prot)	14.12 ± 0.99	13.35 ± 0.20	14.45 ± 0.11	14.01 ± 0.22	14.98 ± 0.48
GCS/(U/mg prot)	9.79 ± 0.07 ^b^	7.89 ± 0.20 ^c^	11.33 ± 0.11 ^ab^	7.84 ± 0.61 ^c^	12.85 ± 0.58 ^a^
GBE/(IU/g prot)	214.54 ± 8.73 ^ab^	212.41 ± 7.83 ^ab^	230.73 ± 4.74 ^ab^	205.32 ± 4.30 ^b^	239.01 ± 7.38 ^a^
LG/(mg/g)	154.54 ± 5.57 ^a^	124.27 ± 6.14 ^b^	149.86 ± 6.79 ^ab^	133.61 ± 5.28 ^ab^	159.92 ± 6.26 ^a^

HK, hexokinase; PFK, phosphofructokinase; PK, pyruvate kinase; CoA, acetyl-CoA; LDH, lactic dehydrogenase; IDH, isocitrate dehydrogenase; MDH, malate dehydrogenase; GPa, glycogen phosphorylase-a; GCS, glycogen synthase; GBE, glycogen branching enzyme; LG, liver glycogen.

## Data Availability

The data presented in this study are available in the main article.

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
