# Peer review of "Effects of Five Dietary Carbohydrate Sources on Growth, Glucose Metabolism, Antioxidant Capacity and Immunity of Largemouth Bass (*Micropterus salmoides*)"

_animals, 2024, doi:10.3390/ani14101492_

Round 1

Reviewer 1 Report

Comments and Suggestions for Authors

This study investigated effects of glucose (GLU), tapioca starch (TS), gelatinized tapioca starch (GTS), potato starch (PS) and gelatinized potato starch (GPS) on growth and physiological responses in juvenile largemouth bass. This study would provide new information for the selective carbohydrate sources in artificial feed and help improve fish health in farmed M. salmoides.  There are some descriptions should be changed as follows:

Comment 1: Line 27-28, There was unclear logic description about “PS and GPS could increase …….expressions”. Authors should change this sentence.

Comment 2: Line 43 and Line 47, after “fish” and “imbalance”, there is only a reference cited and we suggest that more references should be added.

Comment 3: Polysaccharides include generally starch, glycogen, and cellulose. Line 53, Herein, “starch polysaccharides” could be more accurate than “polysaccharides”.

Comment 4: Line 56-60, What is the connection between these three sentences?  Please rewrite them.

Comment 5: Line 62, what kind of fish? Please list specific fish species.

Comment 6: Line 71-72, based on your study, “metabolic and immune capability” should be revised to “glucose metabolism, antioxidant and immune capabilities”.

Comment 7: Line 132-133, Which company or brand of “TEK 8500 VET automatic blood analyzer”?

Comment 8: Table 6 showed activites of key enzyme related to glycolysis and TCA. Why just presented the methods of enzyme related to glycolysis, Line 144? Additionally, in Table 6, “acetyl-CoA” is instead of “CoA”, according to your manuscript.

Comment 9: Line 328-333, please add their Latin names of fish including Nile tilapia, golden pompano, swamp eel, snakehead and large yellow croaker.

Comment 10: Line 429, “bacteria” refers to all bacteria?

Comments on the Quality of English Language

The language of this manuscript needs further improvement.

Author Response

Thank you for your letter and the reviewers’ comments concerning our manuscript entitled “Effects of five dietary carbohydrate sources on growth, glucose metabolism, antioxidant capacity and immunity of largemouth bass (Micropterus salmoides)”, which are valuable and very helpful. We have read through comments carefully and made corrections, according to reviewers’ comments. The responses to the reviewers’ comments are presented following.

Comment 1: Line 27-28, There was unclear logic description about “PS and GPS could increase …….expressions”. Authors should change this sentence.

Response 1: We sincerely thanked the reviewer’s suggestion and comments. This sentence has been revised as “PS and GPS enhanced glycolysis and TCA cycle by increasing their enzyme activities and transcriptional levels” in the new manuscript (Line 28-29).

Comment 2: Line 43 and Line 47, after “fish” and “imbalance”, there is only a reference cited and we suggest that more references should be added.

Response 2: We sincerely thanked the reviewer’s suggestion and comments. According to your suggestion, we have added more references, after “fish” and “imbalance” in the new manuscript (Line 50).

Comment 3: Polysaccharides include generally starch, glycogen, and cellulose. Line 53, Herein, “starch polysaccharides” could be more accurate than “polysaccharides”.

Response 3: We sincerely thanked the reviewer’s suggestion and comments. “polysaccharides” has been changed into  “starch polysaccharides” in the new manuscript (Line 56).

Comment 4: Line 56-60, What is the connection between these three sentences?  Please rewrite them.

Response 4:  We sincerely thanked the reviewer’s suggestion and comments. We have changed original sentence to “And these starch sources have lower amylose content and higher amylopectin content, thereby which are also used as a better mucoadhesive polymer for producing feeds due to a high swelling power” in the new manuscript (Line 59-61).

Comment 5: Line 62, what kind of fish? Please list specific fish species.

Response 5: We sincerely thanked the reviewer’s suggestion and comments. barramundi (Lates calcarifer), and olive flounder (Paralichthys olivaceus) have been listed in the new manuscript (Line 64).

Comment 6: Line 71-72, based on your study, “metabolic and immune capability” should be revised to “glucose metabolism, antioxidant and immune capabilities”.

Response 6: We sincerely thanked the reviewer’s suggestion and comments. According to your advice, it has been revised in the new manuscript (Line 74-75).

Comment 7: Line 132-133, Which company or brand of “TEK 8500 VET automatic blood analyzer”?

Response 7: We sincerely thanked the reviewer’s suggestion and comments. Some information of TEK 8500 VET automatic blood analyzer has been added in the new manuscript (Line 140-141).

Comment 8: Table 6 showed activites of key enzyme related to glycolysis and TCA. Why just presented the methods of enzyme related to glycolysis, Line 144? Additionally, in Table 6, “acetyl-CoA” is instead of “CoA”, according to your manuscript.

Response 8: We sincerely thanked the reviewer’s suggestion and comments. Thanks for your comments. We have revised these mistakes in the new manuscript (Line 153-156, Line 250).

Comment 9: Line 328-333, please add their Latin names of fish including Nile tilapia, golden pompano, swamp eel, snakehead and large yellow croaker.

Response 9: We sincerely thanked the reviewer’s suggestion and comments. We have added their Latin names in the new manuscript (Line 339-346).

Comment 10: Line 429, “bacteria” refers to all bacteria?

Response 10: We sincerely thanked the reviewer’s suggestion and comments.  “bacteria” mainly included Aeromonas hydrophila and Pseudomonas fluorescens. We have revised the corresponding sentence in the new manuscript (Line 450).

Reviewer 2 Report

Comments and Suggestions for Authors

It is a valuable MS in many respects, but needs some amendments and corrections. The M&M part has to give much more details about the experimental conditions as well as the sampling and the calculation of indexes.

An overview of its English by  a native speaker is also recommended. My questions and advice for corrections are given in the attached file.

Comments on the Quality of English Language

An overview of its English by  a native speaker is recommended.

Author Response

Thank you for your letter and the reviewers’ comments concerning our manuscript entitled “Effects of five dietary carbohydrate sources on growth, glucose metabolism, antioxidant capacity and immunity of largemouth bass (Micropterus salmoides)”, which are valuable and very helpful. We have read through comments carefully and made corrections, according to reviewers’ comments. The responses to the reviewers’ comments are presented following.

Comment 1: Line 38, most important/common

Response 1: We sincerely thanked the reviewer’s suggestion and comments. “important” has been added after “most” in the new manuscript (Line 40).

Comment 2: Line 39, used/applied

Response 2: We sincerely thanked the reviewer’s suggestion and comments. Line 41, “supplemented” has been changed to “applied”.

Comment 3: Line 104, 5

Response 3: We sincerely thanked the reviewer’s suggestion and comments. We have revised this mistake to “5” in the new manuscript (Line 108).

Comment 4: Line 104, Water exchange? Cleaning of the tanks?

Response 4: We sincerely thanked the reviewer’s suggestion and comments. This experiment was conducted in a closed circulating system. According to your suggestion, we added this information “in a closed circulating system” in the new manuscript (Line 109).

Comment 5: Line 98-100, From where came he idea of using glucose and it as control? I have never heard of applying it in fish feed as energy source, so please justify that.

Response 5:  We sincerely thanked the reviewer’s suggestion and comments. Although glucose indeed is not used in fish feed, it has been widely used in  studies on carbohydrate sources, for example:

(1)Cui, X.; Zhou, Q.; Liang, H.; Yang, J.; Zhao, L. Effects of dietary carbohydrate sources on the growth performance and hepatic carbohydrate metabolic enzyme activities of juvenile cobia (Rachycentron canadum Linnaeus.). Aquacult Res 2010, 42(1), 99-107.  

(2)Cui, X.; Zhou, Q.; Liang, H.; Yang, J.; Zhao, L. Effects of dietary carbohydrate sources on the growth performance and hepatic carbohydrate metabolic enzyme activities of juvenile cobia (Rachycentron canadum Linnaeus.). Aquacult Res 2010, 42(1), 99-107.  

(3)Deng, D. F.; Hemre, G. I.; Storebakken, T.; Shiau, S. Y.; Hung, S. S. Utilization of diets with hydrolyzed potato starch, or glucose by juvenile white sturgeon (Acipenser transmontanus), as affected by Maillard reaction during feed processing. Aquaculture, 2005, 248(1-4), 103-109.  

(4)Jiang, M.; Liu, W.; Wen, H.; Huang, F.; Wu, F.; Tian, J.; Yang, C.; Wang, W.; Wei, Q. Effect of dietary carbohydrate sources on the growth performance, feed utilization, muscle composition, postprandial glycemic and glycogen response of Amur sturgeon, A cipenser schrenckii Brandt, 1869. J Appl Ichthyol 2014, 30(6), 1613-1619.).

So glucose was set as control, in order to comparatively study effect of structure of carbohydrate sources on growth and metabolism of fish in this study.

Comment 6: In Table 1, mineral

Response 6:  We sincerely thanked the reviewer’s suggestion and comments. In Table 1, “Ineral” has been revised to “Mineral”  in the new manuscript (Line 115).

Comment 7: Line 125, first had to be taken from the fish, not?

Response 7: We sincerely thanked the reviewer’s suggestion and comments. Yes, serum from fish and we have changed “the serum…..10 minutes” to “Blood was taken from the tail vein of fish and then centrifugated at 4°C and 3000 g for a duration of 10 minutes for collecting serum samples”  in the new manuscript (Line 130-134)

Comment 8: Line 135, the serum content of hormones

Response 8: We sincerely thanked the reviewer’s suggestion and comments. Line 135, “the serum content of hormones” was replaced with “the contents of hormones in serum”  in the new manuscript (Line 144-145).

Comment 9: Line 139, determination/measuring

Response 9: We sincerely thanked the reviewer’s suggestion and comments. Line 139, We have changed to determination  in the new manuscript (Line 148).

Comment 10: In Table 4, This was not mentioned in MandM, why? The related text should be put before the tables!

Response 10: We sincerely thanked the reviewer’s suggestion and comments.  In materials and methods, counting methods of blood cell indexes were described in a previous study, and now this reference has been listed  in the new manuscript (Line 139-143).

Comment 11: Line 258, 281, 307, reference for the figures needed.

Response 11: We sincerely thanked the reviewer’s suggestion and comments. In Results section, we have added the reference for these tables and figures in the new manuscript (Line 234, 238, 258, 259, 264, 287, 308, 314, 318).

Comment 12: Line 316, omit thus trivial sentence.

Response 12: We sincerely thanked the reviewer’s suggestion and comments. We have delete this sentence in the new manuscript.

Comment 13: Line 328-329, comparing with….our results indicate…..

Response 13: We sincerely thanked the reviewer’s suggestion and comments. We have revised this sentence in the new manuscript (Line 340-341).

Comment 14: Line 436, better than what?

Response 14: We sincerely thanked the reviewer’s suggestion and comments. Line 436, dietary starch sources (TS, GTS, PS and GPS) exhibited better growth 436 performance, compared with GLU and this sentence has been corrected in the new manuscript (Line 458, 465).

Comment 15: Line 442, would, selective

Response 15: We sincerely thanked the reviewer’s suggestion and comments.  “would” and “selective” were changed to “could” and “selecting”, respectively, in the new manuscript (Line 465).
